# Oncostatin M and STAT3 Signaling Pathways Support Human Trophoblast Differentiation by Inhibiting Inflammatory Stress in Response to IFNγ and GM-CSF

**DOI:** 10.3390/cells13030229

**Published:** 2024-01-25

**Authors:** Marion Ravelojaona, Julie Girouard, Emmanuelle Stella Kana Tsapi, Megan Chambers, Cathy Vaillancourt, Céline Van Themsche, Catherine A. Thornton, Carlos Reyes-Moreno

**Affiliations:** 1Groupe de Recherche en Signalisation Cellulaire (GRSC), Département de Biologie Médicale, Université du Québec à Trois-Rivières, 3351 Boul. des Forges, Trois-Rivières, QC G8Z 4M3, Canada; 2Centre de Recherche Interuniversitaire en Reproduction et Développement-Réseau Québécois en Reproduction (CIRD-RQR), Université de Montréal, St-Hyacinthe, QC J2S 2M2, Canada; cathy.vaillancourt@inrs.ca; 3Regroupement Intersectoriel de Recherche en Santé de l’Université du Québec (RISUQ), Université du Québec, Québec, QC G1K 9H7, Canada; 4Medical School, Swansea University, Swansea SA2 8PP, UK; 5Centre Armand Frappier Santé Biotechnologie, Institut National de la Recherche Scientifique (INRS), Laval, QC H7V 1B7, Canada

**Keywords:** beta-human chorionic gonadotrophin, granulocyte–macrophage colony-stimulating factor, inflammatory stress, interferon gamma, oncostatin M, pregnancy, placenta, syncytiotrophoblast

## Abstract

Interleukin-6 (IL-6) superfamily cytokines play critical roles during human pregnancy by promoting trophoblast differentiation, invasion, and endocrine function, and maintaining embryo immunotolerance and protection. In contrast, the unbalanced activity of pro-inflammatory factors such as interferon gamma (IFNγ) and granulocyte–macrophage colony-stimulating factor (GM-CSF) at the maternal–fetal interface have detrimental effects on trophoblast function and differentiation. This study demonstrates how the IL-6 cytokine family member oncostatin M (OSM) and STAT3 activation regulate trophoblast fusion and endocrine function in response to pro-inflammatory stress induced by IFNγ and GM-CSF. Using human cytotrophoblast-like BeWo (CT/BW) cells, differentiated in villous syncytiotrophoblast (VST/BW) cells, we show that beta-human chorionic gonadotrophin (βhCG) production and cell fusion process are affected in response to IFNγ or GM-CSF. However, those effects are abrogated with OSM by modulating the activation of IFNγ-STAT1 and GM-CSF-STAT5 signaling pathways. OSM stimulation enhances the expression of STAT3, the phosphorylation of STAT3 and SMAD2, and the induction of negative regulators of inflammation (e.g., IL-10 and TGFβ1) and cytokine signaling (e.g., SOCS1 and SOCS3). Using STAT3-deficient VST/BW cells, we show that STAT3 expression is required for OSM to regulate the effects of IFNγ in βhCG and E-cadherin expression. In contrast, OSM retains its modulatory effect on GM-CSF-STAT5 pathway activation even in STAT3-deficient VST/BW cells, suggesting that OSM uses STAT3-dependent and -independent mechanisms to modulate the activation of pro-inflammatory pathways IFNγ-STAT1 and GM-CSF-STAT5. Moreover, STAT3 deficiency in VST/BW cells leads to the production of both a large amount of βhCG and an enhanced expression of activated STAT5 induced by GM-CSF, independently of OSM, suggesting a key role for STAT3 in βhCG production and trophoblast differentiation through STAT5 modulation. In conclusion, our study describes for the first time the critical role played by OSM and STAT3 signaling pathways to preserve and regulate trophoblast biological functions during inflammatory stress.

## 1. Introduction

The placenta has a central role in promoting the homeostatic environment necessary for successful pregnancy [1,2]. Placental structures undergo dynamic changes to coordinate and assure fetal development during pregnancy. Therefore, the placenta is a complex structure that evolves throughout gestation through the differentiation of the fetal trophoblast [1,2]. Trophoblast cells are derived from the trophectoderm through the continuous proliferation of cytotrophoblast (CT) cells and their subsequent differentiation into villous syncytiotrophoblast (VST) and extravillous trophoblast (EVT) cells [3,4]. EVT cells are migratory and invasive mononuclear cells that penetrate the maternal endometrium and anchor the placental villi into the maternal decidua [5]. VST cells form a multinucleated placental surface bathed in the maternal blood and contribute to a successful pregnancy by secreting the hormones necessary for pregnancy, transporting nutrients, mediating gas exchange, balancing immunotolerance, and resisting pathogen infection [6,7]. Aberrations in the formation of VST have been linked to various placental dysfunction syndromes, such as preeclampsia, fetal growth restriction, preterm birth, and stillbirth [6,7]. Therefore, acquiring a comprehensive understanding of the mechanism of syncytium formation will help us further prevent the development of pathological pregnancies.

The principal topics in placental syncytium investigations are fusion and endocrine functions. The placental syncytium is formed by the continuous fusion of the underlying villous cytotrophoblasts [6,7]. The fusion process is led by an enhanced expression of fusogenic proteins such as syncytin-1 [8] and a decreased expression of adhesion molecules such as E-cadherin [9,10]. Two subtypes of the placental syncytium have been identified, the primitive and the definitive syncytium [8]. The primitive syncytium is an invasive type associated with the implanting conceptus characterized by fused trophoblasts, secreting large amounts of human chorionic gonadotrophin (βhCG) and rapidly proliferating to form primary villi [11,12]. The definitive syncytium is formed by the villous syncytiotrophoblast (VST) of the mature placenta [13]. The VST first forms after villi begin to project from the inner cytotrophoblast shell at around the end of two weeks post-coitum. In contrast to the primitive syncytium, the VST forms a single-layered shell over the maternal blood-facing surface of the villi [14]. The VST is also a major endocrine organ, secreting hormones and proteins into the maternal circulation to drive the physiological and metabolic adaptations to pregnancy [15]. Among the various hormones produced by the primitive syncytium and VST, βhCG is the first specific molecule synthesized by the embryo after implantation, stimulating trophoblast invasion and differentiation throughout reaching its peak during the first trimester of pregnancy and gradually decreasing up to the 19th week [16].

During embryo implantation, uterine and placental inflammation represents a normal manifestation of the maternal innate immune system to favor the biological functions of trophoblasts [17,18]. In fact, the innate immune cells and trophoblast cells located at the maternal–fetal interface perform precise cross-talking by producing cytokines, chemokines, and growth factors such as interleukin-10 (IL 10) or transforming growth factor beta (TGFβ) to shape and maintain an immunotolerant environment during pregnancy [19]. Moreover, VST cells release a range of immune signaling molecules to drive the maternal immune response to pathogenic infection during pregnancy through the differentiation and activation of immune cells present at the implantation site [20,21,22,23]. However, excessive inflammation contributes to the pathogenesis of placental inflammatory diseases (chorioamnionitis, placentitis, or villitis), representing a particularly deleterious state for pregnancy outcomes [24,25]. These unbalanced inflammatory conditions enhance the activation of innate immune cells, mainly macrophages and NK cells, with the elevated and sustained expression of pro-inflammatory factors such as GM-CSF and IFNγ, which are widely associated with adverse maternofetal outcomes such as abortion, preterm birth, or preeclampsia [26,27,28,29,30].

Among the various actors involved in immune regulation and trophoblast immune programming within the human pregnant uterus, the interleukin-6 (IL-6) family cytokines such as the leukemia inhibitory factor (LIF) and oncostatin M (OSM) have been reported to support the embryo implantation [31,32] and modulate inflammation [33,34]. Several studies described LIF as a critical factor in preparing the uterine endometrium for implantation by stimulating stromal decidualization and regulating trophoblast cell differentiation and function [32,35,36]. Also, βhCG production stimulates the secretion of LIF and controls IL-6 secretion by endometrial cells to support embryo implantation [9]. In this context, our previous studies described the mediating role of LIF to modulate the pro-inflammatory activity of IFNγ and GM-CSF in macrophages and trophoblast cells by reducing the phosphorylation levels of the signal transducer and activator of transcription-1 (STAT1) and -5 (STAT5). LIF activity induces the activation of anti-inflammatory pathways such as STAT3 and promotes the anti-inflammatory factor IL-10 to preserve trophoblast functions: migration and invasion [37,38]. Then, in another previous study, we described LIF involvement in syncytium formation and βhCG production through STAT3-dependent mechanisms [39]. In this present study, we are interested in OSM, a LIF-stimulated cytokine at the maternal–fetal interface [40]. OSM and LIF are the most pleiotropic members of the IL 6 family cytokines and play significant roles in inflammation [41]. Like LIF within the immune system, OSM has been found to maintain an efficient host response to tissue injury and act as a defense against pathogen infections. It provides protection against lipopolysaccharide-induced endotoxemia by boosting IL-10 and/or suppressing TNFα and IFNγ [42]. Also, in pregnant women, OSM is highly expressed in chorionic tissue and decidua [31]. In the mouse uterus, OSM is mainly expressed in the uterine glandular epithelium and the luminal epithelium during early pregnancy, and in the decidual stromal cells surrounding embryos with the onset of implantation on days 5 to 8 of pregnancy [43]. After that, OSM expression is reduced in the decidual stromal cells, concomitantly with decreasing LIF expression [43]. Notably, OSM production is completely absent in the pregnant uterus of LIF-KO mice; thus, OSM production by the luminal epithelium and the stroma adjacent to the embryo is closely related to LIF production [40]. Moreover, during early pregnancy in mice, OSM expression in the uterus induces IL33 expression and supports decidualization via the STAT3-EGR1 signaling pathway [43]. In human trophoblast and decidual cells, OSM expression has an important role in placental endocrine function by stimulating the release of βhCG and inducing various signaling pathways such as JAK/STAT and ERK1/2 MAP kinases. Through the JAK1 or JAK2 kinase activation, OSM induces STAT1, STAT3, and STAT5 phosphorylation through different tyrosine-dependent mechanisms [31,44]. Through STAT3 activation in human EVT, OSM enhances cell invasion by inducing the expression of the matrix metalloproteinases MMP2 and MMP9, under normoxia or hypoxia environments, and cell proliferation by downregulating E-cadherin expression [45,46,47].

Although many studies strongly suggest an important role for OSM in reproduction and inflammation, the direct role of OSM in modulating trophoblast endocrine and immune functions is poorly defined. Therefore, we investigated the interplay between OSM and STAT3 in the modulation of key trophoblast functions such as fusion and endocrine secretion. For these purposes, we used human trophoblast-like BeWo cells differentiated into VST cells [39] to determine the influence of OSM-STAT3 signaling pathway activation on βhCG production, E-cadherin expression, and the activation of IFNγ-STAT1 and GM-CSF-STAT5 signaling pathways.

## 2. Materials and Methods

Cell culture media, serum, and cell culture reagents were purchased from Wisent (St-Bruno, QC, Canada). Cell culture plates and flasks were obtained from Corning Incorporated (Corning, NY, USA). OSM, IFNγ, and GM-CSF cytokines were purchased from Peprotech (Montreal, QC, Canada). Dimethyl sulfoxide (DMSO), bovine serum albumin (BSA), forskolin, methylthiazolyldiphenyl-tetrazolium bromide (MTT), monoclonal peroxidase-conjugated mouse anti-β-actin antibody, and all electrophoresis grade chemicals were obtained from Sigma Chemical Company (Oakville, ON, Canada). Protease and phosphatase inhibitors cocktail EDTA-Free were obtained from Thermo Fisher Scientific (Rockford, IL, USA). Beta-human chorionic gonadotrophin (βhCG) ELISA kit was purchased from DRG International Inc. (Springfield, NJ, USA). Trizol reagent and PCR primers were obtained from Invitrogen (Burlington, ON, Canada). Taq DNA polymerase and M-MuLV reverse transcriptase were obtained from New England Biolabs (Pickering, ON, Canada). Rabbit polyclonal antibodies targeting phospho-STAT1 (pY701; #9171), STAT1 (#9172), phospho-STAT3 (pY705; #9145), STAT3 (#4904), phospho-STAT5 (pY694; #9359), STAT5 (#9358), phospho-SMAD2 (pS465/467; #3108), and SMAD2 (#5359) were purchased from Cell Signaling Technologies (Danvers, MA, USA), and used at 1:1000 dilution in a phosphate-buffered saline (PBS) solution containing 5% BSA (PBS/5% BSA). The horseradish peroxidase-conjugated goat anti-rabbit IgG was obtained from Bio-Rad Laboratories (Mississauga, ON, Canada) and used at 1:5000 dilution in PBS/5% BSA. The chemiluminescence detection kit (Ultra Science Femto Western Substrate; #CCH365) was purchased from FroggaBio (Concord, ON, Canada). For immunofluorescence analysis, rabbit anti-E-cadherin (#3195) and goat anti-rabbit IgG Fab2 Alexa Fluor^®^ 488-conjugated (#4412) antibodies were used, which were purchased from Cell Signaling Technologies (Danvers, MA, USA).

### 2.1. Cell Culture and Differentiation of Human Trophoblast-like BeWo Cells

The human placental choriocarcinoma BeWo cell line (#CCL-98) was obtained from ATCC (Rockville, MD, USA). Cytotrophoblast-like BeWo (CT/BW) cells were maintained as monolayers in a humidified atmosphere with 5% CO_2_ at 37 °C and were grown in RPMI-1640 cell culture media supplemented with 10% heat-inactivated fetal bovine serum (FBS), 1 mM sodium pyruvate, 10 mM HEPES, and 50 μg/mL gentamicin. To maintain an efficient cell culture workflow, adherent cells were inspected daily under an optical microscope to determine cell health and confluency. For cell passaging (every 3 to 4 days) and subculturing, adherent cells at 75% confluency or less were removed through 0.25% trypsin–EDTA–enzymatic treatment for 5–7 min at 37 °C and then harvested and split into 175 cm^2^ culture flasks or counted and subcultured in 24- or 96-well cell culture plates for further experimental needs. Forskolin (FK)-induced CT/BW cell differentiation into VST (VST/BW) cells was performed as previously described [39]. Briefly, cells at a density of 5.0 × 10^5^ cells/500 μL/well were seeded in a 24-well cell culture plate in 10% FBS-RPMI 1640 cell culture media and incubated overnight. To circumvent any changes in cell behavior during the study, all experiments in this study were restricted to using BeWo cells from passages 8 to 15.

### 2.2. Cell Transfection and Clonal Selection

BeWo cells were transfected with SureSilencing™ control or STAT3 small hairpin RNA (shRNA) expression plasmids, containing the puromycin resistance cassette, according to the manufacturer’s instructions (QIAGEN, Toronto, ON, Canada). The transfection method we used was previously optimized for other plasmids into the BeWo cell line, which resulted in high transfection efficiencies. Briefly, cells were seeded into 24-well plates at 70–90% cell confluency and transfected using Lipofectamine 2000 transfection reagent (Thermo Fisher Scientific, Rockford, IL, USA). To generate stable CT/BW cell clones expressing shRNA, transfected cells were selected using 1 µg/mL puromycin via several passages to produce scrambled (sh-scr) and silencing STAT3 (sh-STAT3) cells. As expected, SureSilencing shRNA plasmids knocked down the expression of the target gene by at least 70% in transfected cells in target shRNA-transfected cells relative to negative control shRNA-transfected cells upon selection for antibiotic resistance. After individual clone selection, the efficacy of sh-STAT3 plasmids in interference with STAT3 was confirmed using Western blotting, and cell viability was evaluated using MTT assays as previously described [37,38,39]. For subsequent experiments, the selected sh-scr and sh-STAT3 CT/BW cell clones were grown in an RPMI-1640 culture medium, supplemented with 10% FBS, in a humidified incubator at 37 °C and 5% CO_2_.

### 2.3. Protein Immunodetection

BeWo cells at a density of 5.0 × 10^5^ cells/500 μL/well were seeded into 24-well plates in 10% FBS-RPMI 1640 cell culture media overnight. Cells were then starved in RPMI-1640 without FBS for 3-4 h and treated with the cytokines of interest. After cytokine stimulation periods, cells were lysed by adding preboiled (95 °C) 1× SDS solubilization buffer containing 1.25 mM Tris-Base pH 6.8, 4% SDS, 10% β-mercaptoethanol, 18% glycerol, 0.03% bromophenol blue, and 2% protease and phosphatase inhibitors. Protein samples were resolved using SDS-PAGE, under reducing conditions and then transferred to the PVDF membrane. Blots were probed with rabbit polyclonal primary antibodies against total (t) or phosphorylated (p) forms of STAT1, STAT3, STAT5, and SMAD2 proteins at 1:1000 dilution at 4 °C overnight. To study E-cadherin expression, blots were probed with rabbit polyclonal primary antibodies against E-cadherin proteins at 1:1000 dilution at 4 °C overnight. Membranes were then incubated with HRP-conjugated goat anti-rabbit IgG Ab at 1:3000 dilution for 1 h at room temperature. Also, β-actin at 1:40,000 dilution was used as a loading control. The detected proteins were visualized using an image analyzer system (Alpha Innotech FluorChem FC2 Imaging System).

### 2.4. RNA Isolation and mRNA Quantification by PCR

BeWo cells were plated at a cell density of 1.0 × 10^6^ cells/2 mL/well into 6-well plates and incubated at 37 °C in RPMI-1640 supplemented with 10% FBS overnight. After cytokine treatment, cells were lysed using TRIzol (In Vitrogen, Montreal, QC, Canada), and cellular RNA was isolated using a Direct-zol RNA MiniPrep Kit (Zymo Research, Burlington, ON, Canada). The mRNA expression profile was evaluated using standard reverse transcription polymerase chain reaction (RT-PCR) as previously described [37,38], with the following primers: human *IL10* gene 5′-ACTTTAAGGGTTACCTGGGTTGC-3′ (S) and 5′-TCACATGCGCCTTGATGTCTG-3′ (AS); human *TGFβ1* gene 5′-CACCCGCGTGCTAATGG-3′ (S) and 5′-ATGCTGTGTGTACTCTGCTTGAACT-3′ (AS); and human *GAPDH* gene 5′-GTCAGTGGTGGACCTGACCT-3′ (S) and 5′-TGAGCTTGACAAAGTGGTCG-3′ (AS).

### 2.5. Detection of βhCG Secretion by ELISA Assay

As previously described [39], cells were plated into 24-well plates at a cell density of 2.0 × 10^5^ cells/500 μl/well and incubated at 37 °C in RPMI-1640 supplemented with 10% FBS overnight. At the end of the treatment period with forskolin alone or combined with the cytokines of interest, the media were harvested, clarified using low-speed centrifugation (500× *g*), and kept at −20 °C until quantification using βhCG ELISA kits as per manufacturer’s instructions (DRG International Inc.). Each assay was performed in triplicate and exhibited consistent reproducibility. sh-STAT3 samples were diluted at a 1:10 ratio, while the sh-scr samples were diluted at a 1:5 ratio, considering the minimum and maximum values of the standard curve.

### 2.6. Statistical Analysis

Each assay was performed in triplicate and exhibited consistent reproducibility. Data were expressed as mean ± SD, and the statistical correlation of data between groups was analyzed via one-way ANOVA followed by Bonferroni post-test using Prism software, version 9.5.0 (Prism 9). The null hypothesis of no difference of means between the two groups was analyzed via Tukey tests, and *p* values of ≤0.05 were considered to indicate statistical significance.

## 3. Results

### 3.1. Interplay of OSM and Pro-Inflammatory Cytokines and Their Effects on Forskolin-Induced E-cadherin Expression and βhCG Secretion

To determine the impact of OSM, IFNγ, and GM-CSF on the endocrine function and fusion process of villous cytotrophoblast cells, CT/BW cells were treated for 48 h with DMSO (control); FK alone; FK combined with OSM, IFNγ, or GM-CSF; or OSM in combination with IFNγ (OSM+IFNγ) or GM-CSF (OSM+GM-CSF) to induce differentiation to VST/BW cells (Figure 1A). E-cadherin (E-cad) is a helpful marker for fusion-competent trophoblast cells as protein expression is downregulated during cellular fusion, and is coincident with enhanced βhCG expression [39,48]. Thus, E-cad expression and βhCG production were, respectively, assessed using Western blotting and ELISA assay. CT/BW cells differentiated with FK alone presented a decrease in E-cad expression (Figure 1B,C,E,F) that was coincident with enhanced βhCG expression compared to control (DMSO) CT/BW cells (Figure 1D,G). CT/BW cells treated with a combination of FK and OSM presented an increase in E-cad expression and a lowered βhCG production (1.52 ± 0.23 IU/mL vs. 0.97 ± 0.12 IU/mL) compared to treatment with FK alone (Figure 1C,D,F,G). Then, in CT/BW cells differentiated with FK combined with IFNγ, we observed a significant increase in E-cad expression concomitant with a significant reduction in βhCG production (1.52 ± 0.23 IU/mL to 0.28 ± 0.04 IU/mL) compared to treatment with FK alone (Figure 1C,D). In contrast, in CT/BW cells treated with FK combined with GM-CSF, we observed a robust decrease in E-cad expression and a significant enhancement of βhCG production (1.52 ± 0.23 IU/mL to 2.70 ± 0.41 IU/mL) compared to treatment with FK alone (Figure 1F,G). However, these opposing effects of IFNγ and GM-CSF on E-cad expression and βhCG secretion were abrogated in the presence of OSM. For instance, E-cad expression decreased in cells treated with OSM+IFNγ compared with IFNγ alone (Figure 1C), while βhCG secretion increased from 0.28 ± 0.04 IU/mL to 1.28 ± 0.15 IU/mL (Figure 1D). In cells treated with OSM+GM-CSF, E-cad expression increased (Figure 1F), and the levels of βhCG secretion decreased from 2.70 ± 0.41 IU/mL to 1.70 ± 0.20 IU/mL (Figure 1G). Collectively, these data suggest that OSM sustains trophoblast morphological and biochemical differentiation by reversing the effects of IFNγ and GM-CSF on E-cad expression and βhCG secretion. This demonstrates the versatility and adaptative effect of OSM to support trophoblast function by inhibiting inflammatory stress in response to IFNγ and GM-CSF.

### 3.2. Regulatory Effects of OSM on IFNγ-STAT1 and GM-CSF-STAT5 Signaling Pathways

We have previously established that LIF influences the behavior of cytokine-activated trophoblast cells and macrophages indirectly via the regulatory action of IL-10 produced by LIF-stimulated VST cells [38] and directly through inhibition of pro-inflammatory IFNγ-STAT1 and GM-CSF-STAT5 signaling pathways in a STAT3-dependent manner [37,38,39]. Therefore, in the present study, the influence of OSM on the expression and activation of pro-inflammatory signaling proteins STAT1 and STAT5 was investigated. Hence, after a 48 h forskolin-induced differentiation period, VST/BW cells were pretreated for 48 h with OSM or PBS and then activated with IFNγ or GM-CSF for 5, 15, 30, and 60 min (Figure 2A). Our data indicate that IFNγ-activated p-STAT1 was significantly reduced in VST/BW cells pretreated with OSM compared to PBS-pretreated cells (Figure 2B,C). The respective percentages of the inhibition of IFNγ-induced p-STAT1 were 44% at t = 5 min, 58% at t = 15 min, 38% at t = 30 min, and 35% at t = 60 min. Similar to IFNγ-induced p-STAT1, GM-CSF-induced p-STAT5 was also inhibited when VST/BW cells were pretreated with OSM (Figure 2D,E), showing a significant inhibition rate at 5 min (64%), 15 min (81%), 30 min (79%), and 60 min (85%) of GM-CSF activation. Further analysis of VST/BW cells treated with OSM+IFNγ and OSM+GM-CSF indicated that OSM pretreatment (48 h) did not affect either t-STAT1 or t-STAT5 (Figure 2B,E). These results strongly suggest that OSM regulates the effects of IFNγ and GM-CSF on the biological response in trophoblast cells through the negative modulation of STAT1 and STAT5 protein phosphorylation.

### 3.3. Impact of OSM in the Activation of Signaling Pathway Proteins and the Expression of Negative Regulators of Cytokine Signaling Pathways

In human trophoblast cells, OSM is known to induce the phosphorylation of different signaling pathway proteins such as STAT1, STAT3, and STAT5 [31,44]. To determine the regulatory mechanisms induced by OSM to modulate and support trophoblast functions, after a 48 h forskolin-induced differentiation period, VST/BW cells were treated with PBS or with 20 ng/mL OSM for relatively short and long periods of stimulation (Figure 3). Then, the activated status of signaling pathway proteins was assessed using Western blotting, and the expression of genes involved in negative signaling pathway regulation was evaluated with RT-PCR. Upon short periods of stimulation with OSM (5, 15, 30, 60, and 90 min), the phosphorylation levels of STAT3, STAT1, and STAT5 proteins were determined in VST/BW cells (Figure 3A,B). Compared to the levels of STAT3 phosphorylation, which were sustained until 60 min and 90 min of stimulation, the levels of p-STAT1 and p-STAT5 increased at 5 min of stimulation with OSM but were rapidly downregulated after 30 min of stimulation (Figure 3B). Moreover, after long periods of stimulation with OSM (1, 6, and 24 h), VST/BW cells also expressed constant and increased levels of phosphorylated, activated STAT3 compared to STAT1 and STAT5 phosphorylation levels, which were downregulated after 1 h of OSM stimulation (Figure 3C,D). Cytokine signal transduction strength is speculated to be influenced by positive/negative regulatory circuitry controlling intracellular signaling [49,50,51]. In human and murine cells, the strong induction of suppressor of cytokine signaling (SOCS) mRNA expression is induced as early as 15–30 min after stimulation with LIF or OSM with a peak at 60–90 min and a continuing decline up to 120 min [52]. Subsequently, we investigated whether OSM could also induce the transcription of *SOCS* genes in VST/BW cells after a short period of OSM stimulation (Figure 3E,F). Indeed, *SOCS1* and *SOCS3* genes were rapidly and significantly induced with a respective peak after 30 min (35-fold) and 60 min (43-fold) of OSM stimulation but, as expected, gradually decreased after 90 min and 120 min of stimulation (Figure 3F). Interestingly, the expression of these negative regulators of cytokine signaling pathways is consistent with the downregulation of activated p-STAT1 and p-STAT5 proteins after short and long periods of stimulation (Figure 3A–D). This strongly suggests that OSM stimulation induces a negative feedback loop on STAT1 and STAT5 activation while supporting STAT3 activation in trophoblast cells.

### 3.4. Requirement of STAT3 for OSM to Modulate the Effects of Pro-Inflammatory Cytokines in Trophoblast Differentiation

Because STAT3 is expected to be the main intracellular regulator required by OSM to safeguard trophoblast cell function and prevent the effects of IFNγ and GM-CSF on E-cad expression and βhCG secretion, we used plasmid vectors to generate stable CT/BW cell clones expressing shRNA mediating the efficient RNA interference targeting of STAT3. After transfection and clone selection, the efficacy of STAT3 interference was confirmed using Western blotting. Stable cell clones were designated as sh-scr (control) and sh-STAT3 (deficient) CT/BW cells (Figure 4A). In selected, stable clones, the silencing of *STAT3* gene expression in sh-STAT3 CT/BW cells resulted in a loss of up to −78% in STAT3 expression compared to the control (Figure 4B). Notably, the viability of transfected sh-STAT3 cells was not affected compared to sh-scr cells (Figure 4C).

### 3.5. Evaluation of Functional Silencing of STAT3 Protein Expression in VST/BW Cells

To confirm that STAT3 expression and phosphorylation are essential cellular mechanisms induced by OSM to regulate the effects of IFNγ and GM-CSF on βhCG production and E-cad expression in fusogenic trophoblast cells, sh-scr and sh-STAT3 CT/BW cells were both treated with DMSO (control); forskolin (FK) alone; FK combined with OSM, IFNγ, or GM-CSF; or OSM in combination with IFNγ (OSM+IFNγ) or GM-CSF (OSM+GM-CSF) to induce differentiation to VST/BW cells (Figure 5). STAT3 protein expression and silencing also impacted E-cad expression in FK-differentiated cells (Figure 5A,B). In general, we found that the expression patterns of E-cad in sh-scr cells (Figure 5A,B) were similar to those determined in parental CT/BW cells (Figure 1A,B), with comparable variations in response to FK and cytokines, alone or in combination. Thus, although sh-scr and sh-STAT3 cells were sensitive to FK treatment by reducing E-cad expression, the reduction was higher in sh-scr cells (−63%) than in sh-STAT3 cells (−19%). In addition, treatment with OSM differently affected E-cad expression in both cell types. In sh-scr cells, treatment with FK and OSM led to an increase in the E-cad expression of approximately +16% (Figure 5A,B). In contrast, in sh-STAT3 cells, OSM stimulation had a nonsignificant effect on E-cad expression compared to FK alone (Figure 5A,B). Single treatments with IFNγ or GM-CSF influenced E-cad expression in FK-differentiated cells. In sh-scr cells, E-cad increase was evaluated at +41% after IFNγ stimulation compared to FK alone, whereas in sh-STAT3 cells exposed to IFNγ alone, the E-cad expression was robustly increased by +108% compared to FK alone (Figure 5B). Interestingly, OSM was able to modulate the effect of IFNγ on E-cad expression in sh-scr cells but not in sh-STAT3 cells. Hence, in sh-scr cells treated with FK and co-stimulated with OSM and IFNγ, the E-cad expression decreased by −82% instead of remaining at +100% as in sh-STAT3 cells co-stimulated with FK, OSM, and IFNγ (Figure 5B). On the other hand, in cells treated with FK and stimulated with GM-CSF, the expression of E-cad decreased to −30% in sh-scr cells and −75% in sh-STAT3 cells (Figure 5B). Notably, these important reductions in the E-cad expression induced by GM-CSF alone were significantly inhibited by OSM co-stimulation, as the relative expression increased by +62% in sh-scr cells and by +58% in sh-STAT3 cells compared to GM-CSF alone (Figure 5B). Moreover, we found that sh-STAT3 cells produced higher amounts of βhCG (9.2 vs. 5.2 IU/mL) than sh-scr cells (Figure 5C). Interestingly, these increased βhCG levels were maintained even after a single stimulation with OSM (9.1 vs. 3.9 IU/mL), IFNγ (9.3 vs. 1.8 IU/mL), or GM-CSF (9.7 vs. 6.9 IU/mL). Secondly, in sh-STAT3 cells, OSM stimulation was unable to abrogate the effects of IFNγ and GM-CSF on βhCG secretion (Figure 5C) compared to sh-scr cells. Overall, these data demonstrate that independently of OSM activity, STAT3 expression and activation orchestrate the signaling pathway mechanisms involved in biochemical and morphological trophoblast differentiation.

### 3.6. OSM Pretreatment Inhibits IFN-STAT1 and GM-CSF-STAT5 Signaling Pathways

Through this experiment, we sought to demonstrate the direct effect of OSM and STAT3 activity on the activation of IFN-STAT1 and GM-CSF-STAT5 signaling pathways. Sh-scr and sh-STAT3 cells were initially pretreated for 48 h with PBS (control) or 20 ng/mL OSM. Then, both types of cells were left unstimulated (t = 0 min) or stimulated with PBS (control) or with either IFNγ or GM-CSF for 15, 30, and 60 min. Western blotting was performed to assess the levels of phosphorylated (p) and total (t) STAT1 (Figure 6A,B) and STAT5 (Figure 6C,D) proteins. In sh-scr cells pretreated with OSM, we observed a reduction in the level of p-STAT1 compared to the control, PBS-pretreated cells (Figure 6A,B). This significant inhibition in OSM-pretreated cell samples was maintained for all the periods of stimulation with IFNγ, with the inhibition rates determined at −82%, −91%, and −69% after 15, 30, and 60 min of stimulation, respectively (Figure 6B). Interestingly, in sh-STAT3 cells, IFNγ stimulation induced similar levels of p-STAT1 in the absence of OSM compared to sh-scr cells. However, after pretreatment with OSM, the levels of p-STAT1 decreased significantly only at t = 15 min (−65%) but were not significantly affected for other periods of stimulation. Thus, at t = 30 min and t = 60 min, the inhibition of p-STAT1 was no longer significant (−5% and an increase in p-STAT1, respectively) compared to PBS-pretreated cell samples (Figure 6B). This loss of IFNγ-p-STAT1 pathway inhibition induced by OSM in sh-STAT3 cells further confirms that OSM modulatory effect on IFNγ activity depends on STAT3 expression and directly modulates the activation of IFNγ-STAT1 signaling pathways. Concerning the response to GM-CSF stimulation, we found that the levels of p-STAT5 were robustly enhanced in sh-STAT3 cells compared to those induced in sh-scr cells (Figure 6C,D). For instance, at t = 15 min, the relative p-STAT5 expression exhibited a seven-fold increase in sh-STAT3 cells compared to sh-scr cells. However, pretreatment with OSM led to a significant decrease in the levels of p-STAT5 for all periods of stimulation with GM-CSF in both sh-scr cells and sh-STAT3 cells. This result strongly supports the idea that OSM uses a different pathway than STAT3 to modulate STAT5 activated by GM-CSF.

### 3.7. OSM Induces Immune Mediators and Anti-Inflammatory Signaling Pathways in VST/BW Cells

The results described above demonstrate that OSM does not prevent the overexpression of E-cad (Figure 5B) or inhibit the phosphorylation of STAT1 induced by IFNγ in sh-STAT3 cells (Figure 6A,B), compared to sh-scr cells. On the other hand, OSM reversed the inhibitory effect of GM-CSF on the expression of E-cad (Figure 5B) and decreased/inhibited the expression of p-STAT5 in both sh-scr and sh-STAT3 cells (Figure 6C,D). Therefore, these two processes would be regulated by OSM via STAT3-dependent and -independent mechanisms. Furthermore, the levels of p-STAT1 and p-STAT5, induced by IFNγ and GM-CSF, respectively, were inhibited when sh-scr cells were pretreated for 48 h with OSM (Figure 6). It was also established that OSM strongly induced the expression of p-STAT3 after 24 h of stimulation, while the levels gradually decreased after 30 min of stimulation with OSM (Figure 3B,D). Importantly, there was no induction of p-STAT1 or p-STAT5 after 24 h of stimulation with OSM (Figure 3B,D). Consequently, we propose that long periods of stimulation with OSM might induce the expression of immune regulators such as IL10 and TGFβ1, which are known to inhibit pro-inflammatory signaling pathways such as IFNγ-STAT1 and GM-CSF-STAT5 via the activation of STAT3 [37,38] and SMAD2 [51,53,54] proteins, respectively. To confirm this hypothesis, we first aimed to determine the expression and phosphorylation levels of STAT3 and SMAD2 proteins in VST/BW cells pretreated with OSM for 48 h and the impact of IFNγ or GM-CSF in these processes. Our results indicated that after a 48 h pretreatment, OSM induced the expression of p-STAT3 and t-STAT3, while the levels of p-STAT3 gradually decreased in response to IFNγ, and those of t-STAT3 remained stable (Figure 7A,C,E). Moreover, comparable results were observed in response to a 48 h pretreatment with OSM and stimulation with GM-CSF for 15 and 60 min (Figure 7 B,D,F). On the other hand, we found that OSM enhanced the expression levels of p-SMAD2 without affecting those of t-SMAD2 after a 48 h pretreatment. Notably, the levels of p-SMAD2 and those of t-SMAD2 remained stable in response to both IFNγ and GM-CSF (Figure 7 A,B,E,G,H,J). Next, we assessed the expression of the immune regulators IL10 and TGFβ1 in OSM-stimulated VST/BW cells. Thus, cells were treated for 6, 12, 24, and 48 h with 20 ng/mL OSM, and then gene expression was assessed using RT-PCR (Figure 8A). Our results indicated that, compared to PBS-treated cells (control), the *IL10* and *TGFβ1* genes were significantly induced in OSM-treated cells, and their expression levels gradually increased from 6 h to 24 h but slowly decreased at 48 h (Figure 8B,C). The patterns of IL10 and TGFβ1 expression were consistent with those of STAT3 and SMAD2 activation induced upon long periods of OSM stimulation in VST/BW cells. Collectively, these observations demonstrate that OSM treatment in trophoblast cells enhances anti-inflammatory cytokine production such as IL-10 and TGFβ1, which might be involved in the modulation of IFNγ-STAT1 and GM-CSF-STAT5 signaling pathways. Finally, to confirm that SMAD2 and TGFβ1 expressions were indirectly regulated by OSM treatment through STAT3 activation, sh-scr and sh-STAT3 VST/BW cells were treated for 24 h with PBS or 20 ng/mL OSM, and then STAT3 and SMAD2 activation was assessed using Western blotting (Figure 9A). In sh-scr VST/BW cells, the levels of p-STAT3, t-STAT3, and p-SMAD2 were significantly increased with OSM treatment (Figure 9A–D), while the expression levels of t-SMAD2 remained unchanged compared to PBS-treated cells (Figure 9E). As expected, in sh-STAT3 VST/BW cells, the expression levels of p-STAT3 and t-STAT3 were significantly reduced with similar reduction rates (−61.2%) to those of sh-scr cells (Figure 9B,C). Interestingly, concurrent with this reduction in STAT3 expression in sh-STAT3 cells, we observed a significant reduction in p-SMAD2 expression (−60.8%) compared to sh-scr cells (Figure 9D). In contrast, the expression levels of t-SMAD2 remained unchanged in both PBS- and OSM-treated sh-STAT3 cells (Figure 9E). Thus, these results strongly suggest that STAT3 is the main cellular mechanism induced by OSM to regulate the expression of immune regulatory factors and anti-inflammatory pathways in VST/BW cells.

## 4. Discussion

It is well established that an unbalanced immunological response at the maternal–fetal interface can result in pregnancy disorders such as preterm birth, spontaneous abortion, or preeclampsia [24,25,55]. Aberrant maternal inflammation is known to affect trophoblast cell motility, survival, and function, potentially leading to negative consequences for pregnancy outcomes [26,27,28,29,30]. In this context, our study aimed to establish whether and how the gestational cytokine OSM and the transcriptional factor STAT3 regulated IFNγ- or GM-CSF-mediated endocrine and immunological functions in a model of villous syncytiotrophoblast cells (VST/BW cells).

Although the effects of OSM activity on trophoblast cell migration and proliferation have been previously published using EVT cell models, such as HTR-8/SVneo, JEG-3, or JAR cells [31,45,46,47], the impact of OSM on trophoblast syncytialization and endocrine capacities have not been fully investigated. Among the available trophoblast-like cell lines used as models to investigate placental function, fusogenic BeWo cells are commonly dedicated to studying syncytialization, adhesion, and endocrine function, while non-fusogenic JAR or JEG-3 cell lines are widely chosen to study the molecular mechanisms underlying the proliferation and invasive potential of trophoblast cells [56].

During differentiation, VST cells must undergo morphological and biochemical changes characterized by the syncytialization and production of hormones, such as βhCG and progesterone [57]. The production of placental βhCG is crucial for the maintenance of pregnancy by promoting trophoblast differentiation and immunomodulatory functions [58] by reducing the expression of pro-inflammatory factors [16]. Indeed, a higher expression of pro-inflammatory factors such as TNFα is known to affect trophoblast cell fusion by reducing βhCG expression [59]. In this setting, we described the effects of IFNγ and GM-CSF on morphological and biochemical changes during forskolin-induced VST cell differentiation.

GM-CSF, via STAT5 activation, plays essential roles in placental morphogenesis and fetal development by acting as a maternal immune tolerant factor as well as a trophic growth and viability factor in preimplantation embryos [60,61]. In fact, during early pregnancy, uterine GM-CSF expression remains high for a few days after conception but declines around the time of embryo implantation under the inhibitory influence of progesterone [62]. However, GM-CSF expression or activity at levels above or below the threshold at the maternal–fetal interface induces pregnancy complications such as preeclampsia (PE) and fetal growth restriction (FGR) related to insufficient placental development and function [63].

Depending on the levels secreted in the uterus during early pregnancy, IFNγ also can be either beneficial or detrimental to the outcome of pregnancy [64]. In the immune system, IFNγ is crucial for the stimulation of adaptive immune responses against pathogens [65]. As a gestational factor, IFNγ released by uterine NK cells is involved in inhibiting excessive trophoblast cell invasion into the uterine wall during implantation [66]. Although IFNγ is a necessary cytokine in the establishment and maintenance of pregnancy, the elevated expression or activity levels of IFNγ in uterine decidual and placental tissues are associated with aberrant inflammatory responses, placental dysfunction, and increased risks of abortion and PE [26,27]. Importantly, the molecular process underlying inflammation-related events in PE was established by describing the response of JEG-3 cells to pro-inflammatory cytokines, including IFNγ and GM-CSF, resulting in the inhibition of trophoblast invasion by downregulating the production of matrix metalloproteinases MMP2 and MMP9 [67]. In this context, our previous studies provide further evidence that human trophoblast and macrophage functions are affected by exposure to IFNγ [37,38]. Aiming to investigate the interplay between LIF and trophoblast cells in the modulation of macrophage functional phenotype, we demonstrated that LIF-differentiated VST cells produced IL10 via sustained STAT3 activation to generate IFNγ-activated macrophages with reduced TNFα expression and cytotoxic function [38]. In addition, we also showed that cell activation with IFNγ inhibited cell invasion and migration, but this immobilizing effect in macrophages and trophoblast cells was abrogated by LIF through the induction of STAT3 activation and MMP9 expression [37].

Here, we demonstrated that IFNγ significantly reduced βhCG levels, while GM-CSF substantially upregulated them in VST/BW cells. Notably, abnormally low (<0.5 μM) or high (>2.0 μM) free βhCG levels are generally associated with an increased risk of adverse pregnancy outcomes, i.e., FGR, spontaneous abortion, and preterm birth [68]. Moreover, high total hCG concentration in early pregnancy is associated with an increased risk of PE [69]. Moreover, clinical studies have indicated that abnormal high serum GM-CSF levels are strongly associated with elevated serum βhCG levels in progressive tumor cases of choriocarcinoma [70], suggesting a potential pathological role for higher levels of GM-CSF and βhCG in gestational trophoblastic diseases. The upregulation of βhCG by STAT5 signaling inducers, such as GM-CSF, is consistent with previous studies using BeWo cells as a VST model indicating that FK activates the cAMP-PKA signaling pathway, which in turn upregulates STAT5 signaling activity, resulting in enhanced βhCG production [57]. Interestingly, our results demonstrate that OSM counteracts the effects of IFNγ and GM-CSF on trophoblast hormonal function by nearly restoring the increased or decreased levels of βhCG to those normally induced in VST/BW cells after differentiation with forskolin alone. E-cadherin is a cell–cell adhesion molecule normally expressed in villous trophoblasts, but it diminishes when they differentiate into the VST of the first- and second-trimester placenta [71]. Indeed, as described during the in vitro VST differentiation of human chorionic villi trophoblasts and BeWo cells, E-cadherin levels are significantly increased during earlier stages of syncytialization but rapidly decreased as syncytialization progresses in the spontaneous and induced syncytialization systems [72]. Although the literature regarding the relationship between E-cadherin expression and gestational pathologies is controversial, it is predominantly in the direction of an increased expression of E-cadherin in PE and FGR [73,74]. In this context, we found that IFNγ and GM-CSF also had opposing effects on morphologic trophoblast differentiation, with IFNγ increasing E-cadherin expression, whereas GM-CSF reduced it in VST/BW cells. However, we found that OSM stimulation efficiently corrected the effect of IFNγ and GM-CSF on morphologic differentiation of trophoblast cells by restoring the increased or decreased levels of E-cadherin expression induced by pro-inflammatory cytokines.

Together, these data strongly suggest that functional OSM signaling effectively modulates pro-inflammatory signal transduction in trophoblast cells, and we demonstrated that OSM negatively regulated IFNγ-STAT1 and GM-CSF-STAT5 signaling pathway activation. Interestingly, the OSM inhibitory effect was mirrored by increased total and phosphorylated STAT3 protein levels and accompanied by a significant enhancement of SMAD2 protein phosphorylation. We established that OSM induces the expression of genes encoding for IL10, TGFβ1, SOCS1, and SOCS3 proteins in trophoblast cells. Notably, the immunomodulatory cytokine IL10 is known to inhibit IFNγ-STAT1 activation and STAT1-dependent gene expression through the STAT3-dependent expression of *SOCS1* and *SOCS3* genes [75,76,77]. In contrast, the molecular mechanisms through which TGFβ1 inhibits IFNγ-STAT1 are mainly dependent on SMAD2/3 pathway activation through the recruitment of SOCS1 [53] and/or PIAS1 (a protein inhibitor of activated STAT1) [54] or the TGFβ receptor 1 (TGFβR1)-mediated phosphorylation of the IFNγ receptor 1 (IFNGR1) [51]. Moreover, SOCS1 also negatively regulates GM-CSF signaling by ubiquitylating the GM-CSF receptor (GMR) for subsequent destruction via the proteasome [78]. Thus, mechanistically, it is conceivable to propose the OSM modulation of IFNγ-STAT1 and GM-CSF-STAT5 signaling pathways in trophoblast cells through different mechanisms, including the expression of IL10 and TGFβ1, the activation of STAT3 and SMAD2, and the recruitment of SOCS1 and SOCS3 proteins (Figure 10).

STAT3 expression is essential for the early development of the embryo, and STAT3 knockout mice undergo a notable loss of fertility due to embryonic lethality in early gestation [79]. Moreover, activated STAT3 is proposed to be a critical modulator of trophoblast invasion, as the inhibition of STAT3 phosphorylation in the mouse endometrium also prevents embryo implantation [80]. However, the role of STAT3 in the human trophoblast differentiation process is not fully understood. Studies have reported that the spontaneous and induced differentiation of trophoblast cells is strongly associated with increased STAT3 expression [81]. In addition, phosphorylated STAT3 protein detection is associated with normal villous and extravillous trophoblast cells of the human first-trimester placenta; afterward, this expression profile disappears in the term placenta [82]. Thus, this observation strongly suggests that STAT3 may play a significant role in the induction and maintenance of trophoblast differentiation and function in early pregnancy.

To further understand the impact of OSM and STAT3 on trophoblast functions such as fusion and endocrine secretion, loss-of-function studies were performed using STAT3-deficient BeWo cells. The most impressive effect of STAT3 inactivation during BeWo cell differentiation is the overproduction of βhCG compared to levels from STAT3-expressing VST/BW cells. Our data are consistent with previous findings reporting a significant increase in βhCG secretion in transient STAT3-deficient BeWo cells differentiated to VST cells [81]. These findings have potential biological and clinical significance because abnormally increased βhCG expression [83] with decreased STAT3 expression has been reported in placental chorionic villi from idiopathic FGR pregnancies [81]. Thus, our results support the idea that decreased expression or activation of STAT3 may lead to the upregulation of βhCG, which in turn could prematurely deplete the pool of proliferating trophoblast cells by favoring terminal trophoblast differentiation and syncytialization in FGR.

Another major discovery is that functional cytokine signaling is impaired in STAT3-deficient cells. For instance, compared to STAT3-expressing VST/BW cells, stable, higher levels of βhCG were maintained in STAT3-deficient VST/BW cells even after stimulation with OSM, IFNγ, or GM-CSF individually, or with OSM combined with IFNγ or GM-CSF. STAT3 deficiency in VST/BW cells also resulted in enhanced E-cadherin expression levels compared to those normally expressed in STAT3-expressing VST/BW cells. In contrast to βhCG, the regulation of E-cadherin expression was maintained to some extent in STAT3-deficient VST/BW cells, as the expression levels were increased in response to OSM alone and even more when stimulated with IFNγ alone but strongly reduced in response to GM-CSF alone compared to STAT3-expressing VST/BW cells. In combination, OSM stimulation did not reduce enhanced IFNγ-mediated E-cadherin expression but significantly increased the expression of GM-CSF-mediated diminished E-cadherin. Notably, STAT3-deficient VST/BW cells were unable to negatively control the activation of both IFNγ-STAT1 signaling pathways compared to GM-CSF-STAT5 in response to OSM. This suggests that OSM uses different pathways to modulate inflammatory stress signaling in trophoblast cells. In addition, STAT3-deficient VST/BW cells showed an enhanced expression of phosphorylated STAT5 but a reduced expression of E-cadherin in response to GM-CSF compared to STAT3-expressing VST/BW cells. These results are in line with those indicating that the STAT5 signaling pathway is upregulated during forskolin-induced VST differentiation, resulting in enhanced βhCG production and reduced E-cadherin expression [57].

## 5. Conclusions

Collectively, these data suggest a dependence of OSM for STAT3 in the regulation of βhCG but not E-cadherin expression, as well as in the regulation of signal transduction in response to activated IFNγ-STAT1 signaling pathway, but they also highlight the ability of OSM to regulate the GM-CSF-STAT5 signaling pathway through a STAT3-independent mechanism (Figure 10). Thus, we propose OSM-induced STAT3 activation as a main molecular mechanism for the regulation of IFNγ-STAT1 and GM-CSF-STAT5 signaling pathways’ activation in trophoblast cells. Other mechanisms may involve TGFβ1-SMAD2 signaling pathway activation, and/or the recruitment of SOCS1 and SOCS3 proteins; however, this requires further investigation.

These conclusions are based on in vitro experiments with a tumoral cell line (choriocarcinoma), which presents a limitation in the conclusion of this study. However, the immortalized BeWo cell line appears to fulfill many of the necessary conditions for studying the function of the human placental syncytium and is largely used to study the most pertinent aspects of trophoblast biology [56], often due to ethical restrictions and limited availability of human tissues. Nevertheless, the use of human trophoblast stem cells or placenta explants could be considered for future in vitro investigations. Using trophoblast stem cells would be interesting to explore further fundamental biological effects and the role of OSM in the different trophoblast subtypes. Nonetheless, in the future, we will ascertain whether these OSM mechanisms operate in an in vivo model.

## Figures and Tables

**Figure 1 cells-13-00229-f001:**
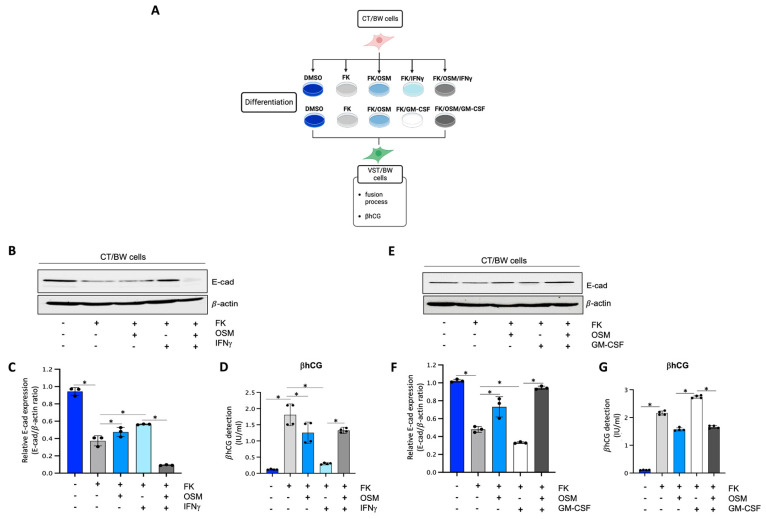
Influence of OSM, IFNγ, and GM-CSF on E-cadherin expression and βhCG production in forskolin-differentiated VST/BW cells: (**A**) CT/BW cells were treated for 48 h with 0.2% DMSO as a vehicle (control) or differentiated with 10 μM forskolin (FK) alone; FK combined with OSM (20 ng/mL), IFNγ (5 ng/mL), or GM-CSF (5 ng/mL); or OSM in combination with IFNγ or GM-CSF. Then, βhCG production was measured using ELISA and E-cadherin (E-cad) expression with Western blotting. (**B**–**D**) Representative images and/or graphical analysis of relative expression of E-cad and βhCG production in response to OSM alone or in combination with IFNγ. (**E**–**G**) Representative images and/or graphical analysis of relative expression of E-cad and βhCG production in response to OSM alone or in combination with GM-CSF. In panels (**C**,**F**), the E-cad/β-actin ratios were determined to express data as relative expression of E-cad at each treatment. In panels (**D**,**G**), data are expressed as the number of international units (IU) of βhCG per mL of supernatant. Each bar represents the mean ± SD of three independent experiments; * *p* < 0.05 denotes a significant difference between the cell groups.

**Figure 2 cells-13-00229-f002:**
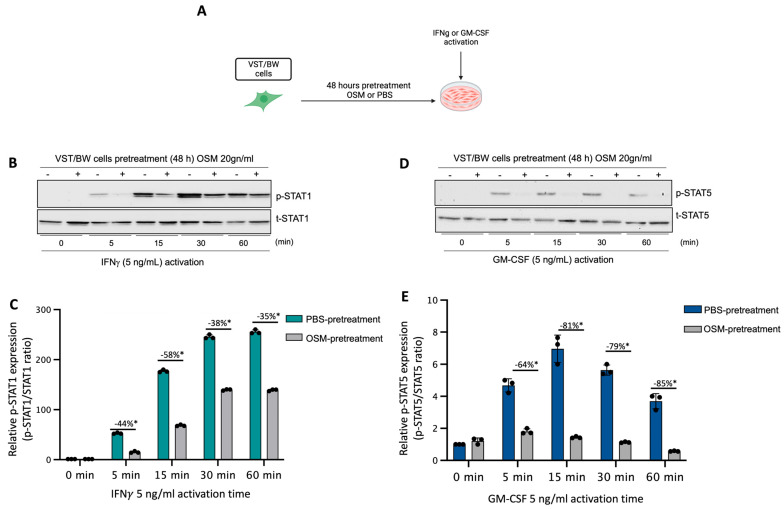
Impact of OSM on the activation of IFNγ-STAT1 and GM-CSF-STAT5 signaling pathways in VST/BW cells: (**A**) VST/BW cells were pretreated for 48 h with PBS as a vehicle (control) or with OSM (20 ng/mL). Then, cells were stimulated with PBS or with either IFNγ (5 ng/mL) or GM-CSF (5 ng/mL) for 5, 15, 30, and 60 min. The relative levels of phosphorylated (p) and total (t) proteins were assessed using Western blotting. (**B**,**C**) Representative images and graphical analysis of relative expression of p-STAT1. (**D**,**E**) Representative images and graphical analysis of relative expression of p-STAT5. The p-STAT1/t-STAT, and p-STAT5/t-STAT5 ratios were determined to express data as the relative expression of p-STAT1 and p-STAT5, respectively, at each stimulation time with IFNγ or GM-CSF. Each bar represents the mean ± SD of three independent experiments; * *p* < 0.05 denotes a significant difference between the impact of OSM at each stimulation time.

**Figure 3 cells-13-00229-f003:**
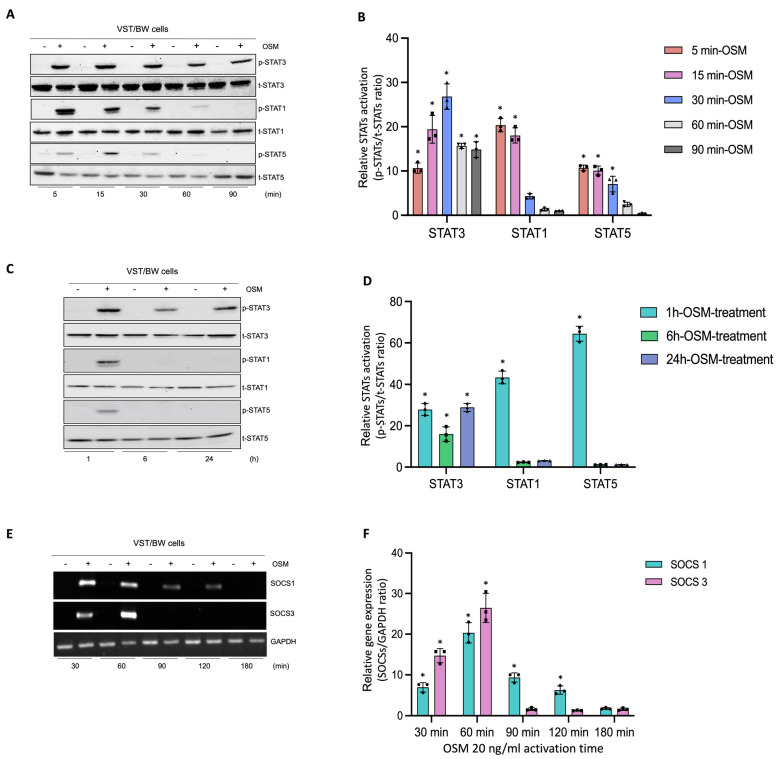
Analysis of protein phosphorylation of STAT1, STAT3, and STAT5, and gene expression of *SOCS1* and *SOCS3* in OSM-stimulated VST/BW cells: (**A**,**C**) Representative images of phosphorylated (p) and total (t) STAT3, STAT1, and STAT5 proteins as assessed using Western blotting, and (**B**,**D**) graphical analysis of relative expression of p-STAT3, p-STAT1, and p-STAT5. Cells were treated for 5, 15, 30, 60, and 90 min (short periods) and 1, 6, and 24 h (long periods) with either PBS as a vehicle (control) or OSM (20 ng/mL). The p-STAT3/t-STAT3, p-STAT1/t-STAT1, and p-STAT5/t-STAT5 ratios were determined to express data as relative expression of p-STAT3, p-STAT1, and p-STAT5, respectively, at each stimulation time. Each bar represents the mean ± SD of three independent experiments; * *p* < 0.05 denotes a significant difference between the impact of OSM at each stimulation time. (**E**) Representative images of *SOCS1* and *SOCS3* gene expression assessed via PCR and (**F**) graphical analysis induction of genes encoding *SOCS1* and *SOCS3* relative to *GAPDH* gene in VST/BW cells stimulated with PBS (control) or with OSM for 30, 60, 90, 120, and 180 min. The *SOCS1/GAPDH* and *SOCS3/GAPDH* ratios were determined to express data as the relative expression of *SOCS1* and SOCS3, respectively, at each stimulation time. Each bar represents the mean ± SD of three independent experiments; * *p* < 0.05 denotes a significant difference compared to the control.

**Figure 4 cells-13-00229-f004:**
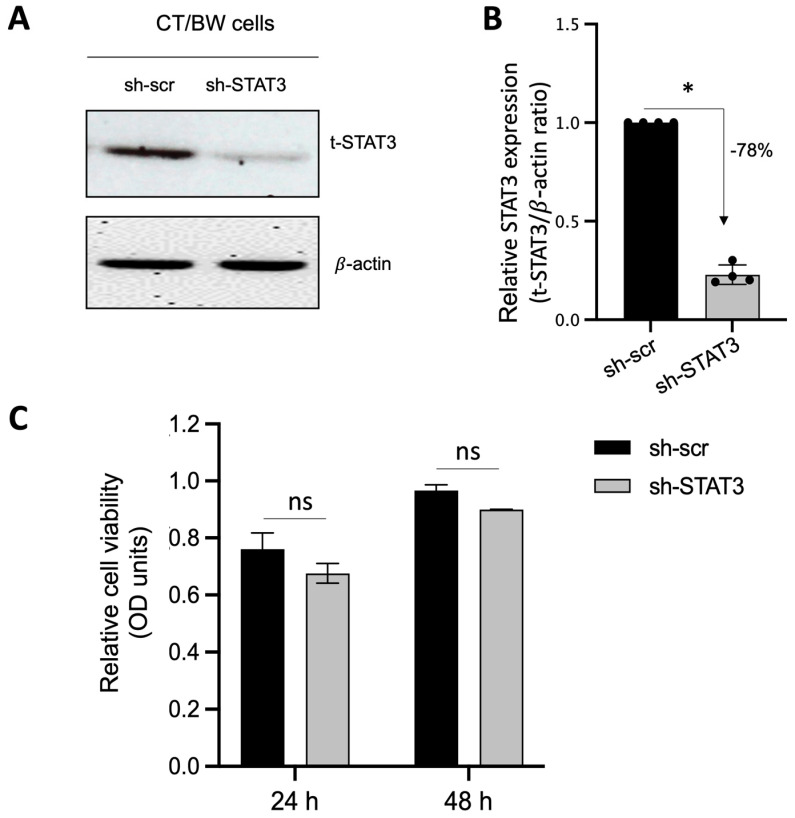
Effectiveness of STAT3 silencing in viable STAT3-deficient BeWo cells (sh-STAT3 CT/BW cells). (**A**,**B**) Representative images and graphical analysis of relative expression of total STAT3 (t-STAT3) in selected CT/BW cell clones stably expressing small hairpin (sh) RNA against control scrambled (sh-scr) mRNA or STAT3 mRNA (sh-STAT3) as evaluated with Western blotting. The t-STAT3/β-actin ratio was determined to express data as the relative expression of STAT3. (**C**) Graphical analysis of relative cell viability assessed using MTT assays in selected sh-scr and sh-STAT3 CT/BW cells. Each bar represents the mean ± SD of four independent experiments; * *p* < 0.05 denotes a significant difference between STAT3 expression in sh-scr CT/BW cells and sh-STAT3 CT/BW cells.

**Figure 5 cells-13-00229-f005:**
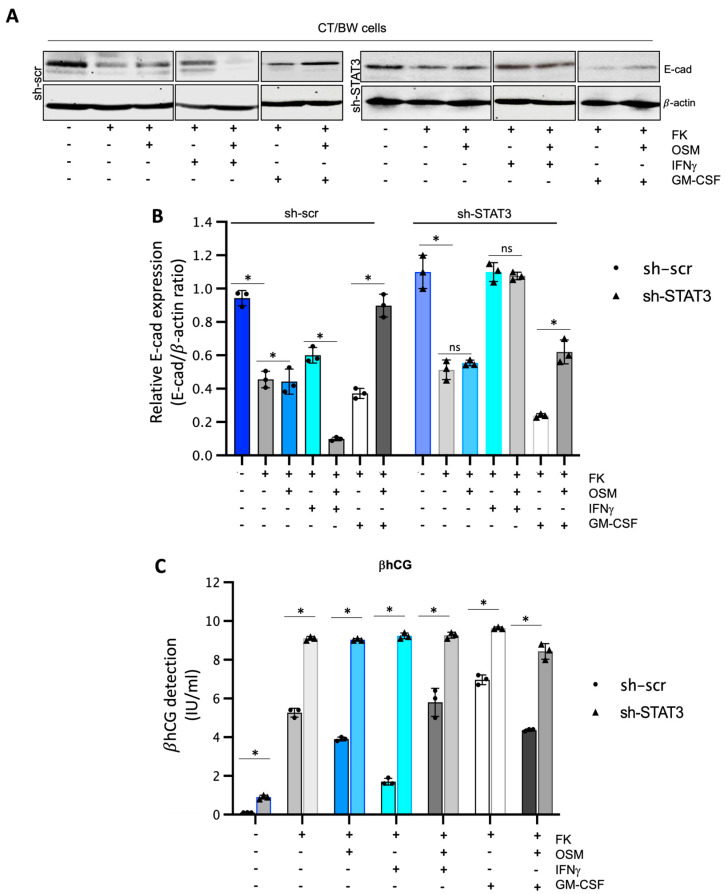
Impact of STAT3 silencing in OSM regulation on IFNγ and GM-CSF effects influencing βhCG secretion and E-cadherin expression in VST/BW cells: (**A**,**B**) Representative images and graphical analysis of relative E-cadherin (E-cad) expression in sh-scr and sh-STAT3 TB/BW cells, treated as indicated above, using Western blotting. The E-cad/β-actin ratio was determined to express data as the relative expression of E-cad. (**C**) Graphical analysis of βhCG production in sh-scr and sh-STAT3 CT/BW cells treated for 48 h with 0.1% DMSO as a vehicle (control); forskolin (10 μM), OSM (20 ng/mL), IFNγ (5 ng/mL), or GM-CSF (5 ng/mL) alone; or OSM in combination with IFNγ or GM-CSF. ELISA assay measured βhCG production in cell-free supernatant (values in quadruplicate). Data are expressed as the number of international units (IU) of βhCG per mL of supernatant. Each bar represents the mean ± SD of three independent experiments. Different superscripts denote significant differences between the samples (* *p* < 0.05) and ns = nonsignificant difference.

**Figure 6 cells-13-00229-f006:**
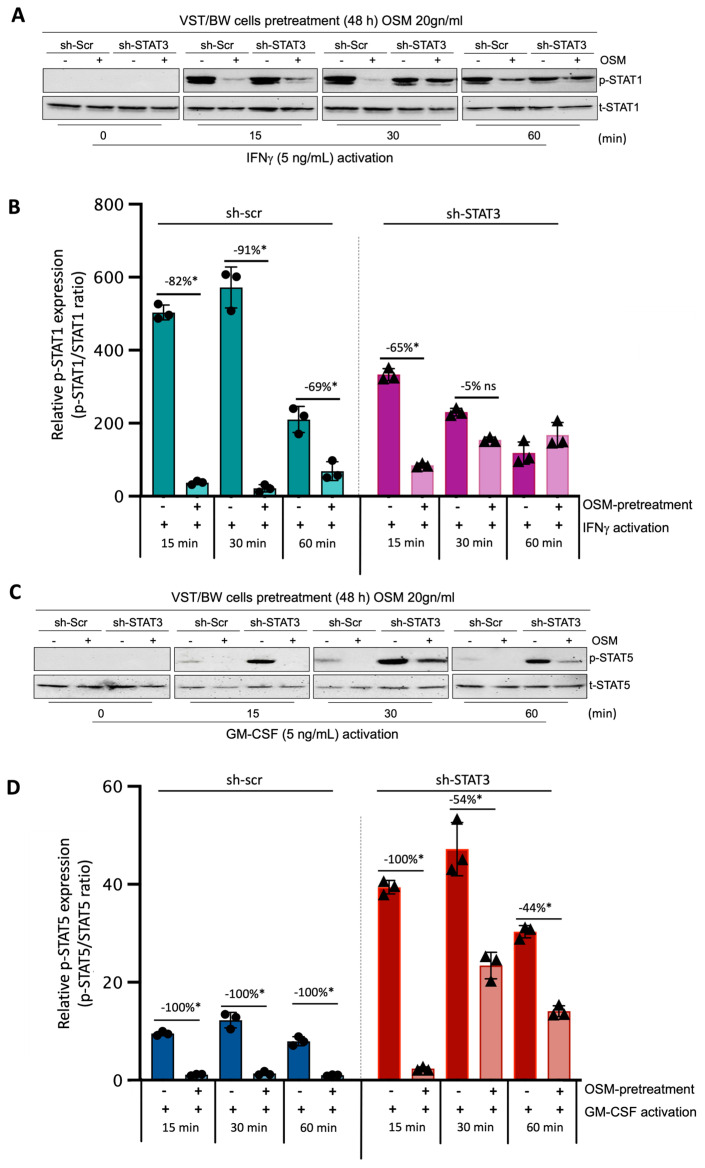
Regulatory effects of OSM on STAT1 and STAT5 protein phosphorylation in IFNγ- and GM-CSF-stimulated sh-scr and sh-STAT3 cells: (**A**,**C**) Representative images of phosphorylated (p) and total (t) STAT1 and STAT5 proteins and graphical analysis of relative expression of (**B**) p-STAT1 and (**D**) p-STAT5. Cells were pretreated for 48 h with either PBS as a vehicle (control) or OSM (20 ng/mL), and then with PBS (t = 0 min), 5 ng/mL IFNγ, or 5 ng/mL GM-CSF for different periods. The p-STAT1/t-STAT1 and p-STAT5/t-STAT5 ratios were determined to express data as the relative expression of p-STAT1 and p-STAT5, respectively, at each stimulation time. Each bar represents the mean ± SD of three independent experiments; * *p* < 0.05 denotes a significant difference between the impact of pretreatment with PBS versus OSM at each stimulation time; ns = non-significant difference.

**Figure 7 cells-13-00229-f007:**
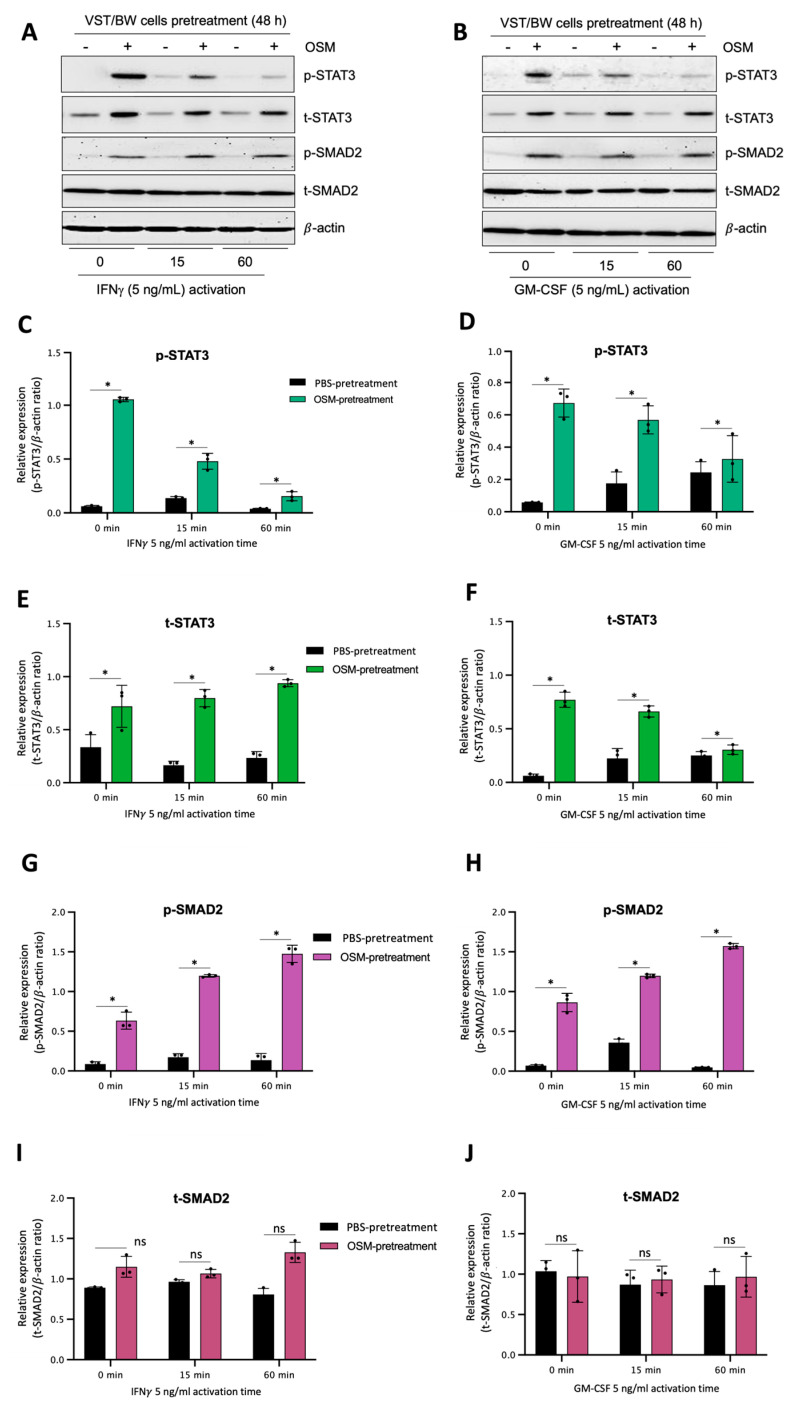
OSM stimulation regulates STAT3 and SMAD2 protein expression and phosphorylation in VST/BW cells stimulated with IFNγ or GM-CSF: (**A**,**B**) Representative images of phosphorylated (p) and total (t) STAT3, SMAD2, STAT1, and STAT5 proteins, and graphical analysis of relative expression of p-STAT3, p-SMAD2, p-STAT1, and p-STAT5 in VST/BW cells pretreated with PBS (control) or with OSM (20 ng/mL) for 48 h, and then activated with IFNγ (**C**,**E**,**G**,**I**) or GM-CSF (**D**,**F**,**H**,**J**) for 15 and 60 min. The phosphorylated (p)/total (t) protein ratio was determined to express data as the relative expression of phosphorylated proteins; * *p* < 0.05 denotes a significant difference between PBS versus OSM pretreatment on protein activation, ns = non-significant difference.

**Figure 8 cells-13-00229-f008:**
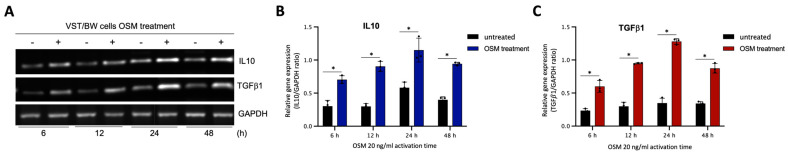
OSM stimulation induces *IL10* and *TGFβ1* gene expression in VST/BW cells: (**A**) Representative images of *IL10* and *TGFβ1* gene expression and (**B**,**C**) graphical analysis of relative expression of genes encoding *IL10* and *TGFβ1* relative to *GAPDH* gene in VST/BW cells stimulated with PBS (control) or with 20 ng/mL OSM for 6, 12, 24, and 48 h. The *IL10/GAPDH* or *TGFβ1/GAPDH* ratio was determined to express data as the relative expression of genes. Each bar represents the mean ± SD of three independent experiments; * *p* < 0.05 denotes a significant difference between the impact of PBS versus OSM stimulation on gene expression.

**Figure 9 cells-13-00229-f009:**
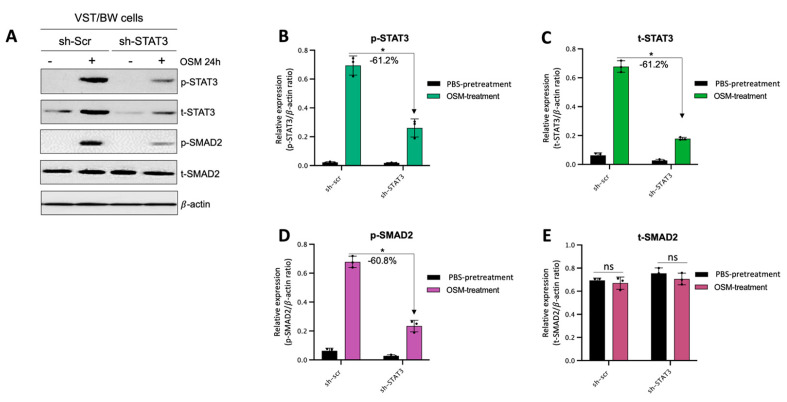
Contribution of STAT3 to OSM regulation of SMAD2 phosphorylation in sh-scr compared to sh-STAT3 VST/BW cells: (**A**) Representative image of phosphorylated (p) and total (t) STAT3, and SMAD2 proteins and graphical analysis of relative expression of (**B**) p-STAT3, (**C**) t-STAT3, (**D**) p-SMAD2, and (**E**) t-SMAD2 in sh-scr and sh-STAT3 VST/BW cells. Cells were treated for 24 h with either PBS (control) or OSM (20 ng/mL). The p-STAT3/β-actin, t-STAT3/β-actin, p-SMAD2/β-actin, and t-SMAD2/β-actin ratios were determined to express data as the relative expression of p-STAT3, t-STAT3, p-SMAD2, and t-SMAD2, respectively, at each stimulation time. Each bar represents the mean ± SD of three independent experiments; * *p* < 0.05 denotes a significant difference between sh-scr and sh-STAT3 VST/BW pretreated with OSM.

**Figure 10 cells-13-00229-f010:**
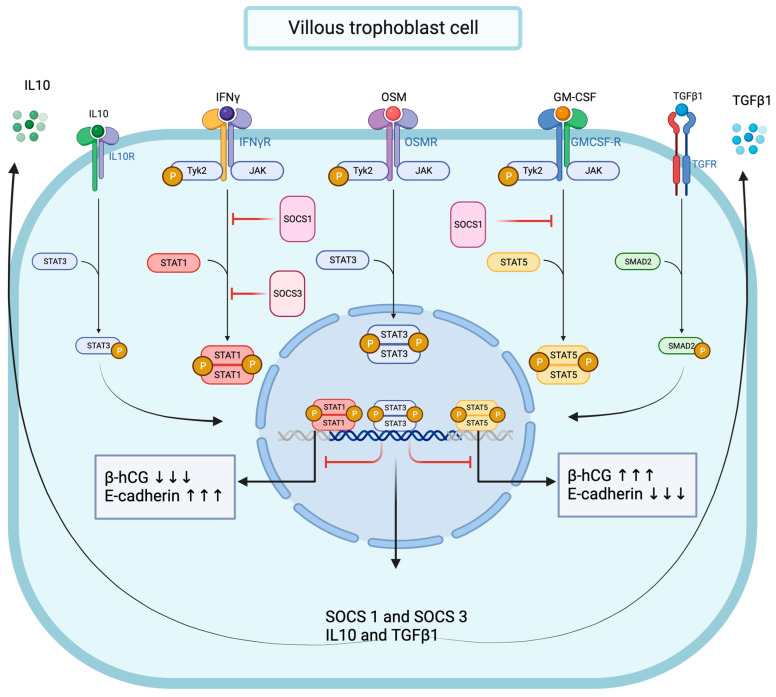
Graphical conclusion of the modulating effects of OSM and STAT3 signaling pathways on IFNγ-STAT1 and GM-CSF-STAT5 signaling pathways’ activation in trophoblast cells.

## Data Availability

The raw data supporting the conclusions of this article will be made available by the authors on request.

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
