# Peer review of "Oncostatin M and STAT3 Signaling Pathways Support Human Trophoblast Differentiation by Inhibiting Inflammatory Stress in Response to IFNγ and GM-CSF"

_cells, 2024, doi:10.3390/cells13030229_

Round 1

Reviewer 1 Report

Comments and Suggestions for Authors

This manuscript by Ravelojaona et al investigates the role of the cytokine Oncostatin M on trophoblast behaviour using the BeWo cell line as a model for syncytiotrophoblast formation. Interactions with IFNg and GM-CSF are being tested, as well as the downstream signaling pathways that are involved. The data appear solid, provide a significant level of detail and are overall interesting.

Issues that need to be fixed:

Line 86: TGFb should be transforming growth factor beta

Line 119-122: It is unlcear where Oncostatin M is expressed. Please clarify and specify between sites of production in the endometrium and in the emerging placenta. ‘Luminal epithelium’ would imply uterus but with progressing development no LE is left, so the source should vanish and therefore not influence syncytiotrophoblast formation, at least not in the mouse to which these data refer. The next sentence refers to OSM production by the placenta but no specific cell type is mentioned as source.

Fig. 1 C vs D, the effect of OSM + IFNg on E-Cadherin expression vs hCG production are confusing, i.e. if the combined action of OSM+IFNg leaves so little E-Cadherin (i.e. cytotrophoblast-like cells) one would expect massive amounts of b-hCG production, well over the FK control. Please explain this discrepancy.

Also, please amend the legend as some bars seem to have 4 data points, and the statistics bars in Fig. 1G are mis-aligned.

Sections 3.2 and 3.3, please explain in more detail the layout of this experiment. Cells were first induced to differentiate into VST/BW by FK treatment and then ‘pre-treated’ with PBS or OSM? How long was the FK-induced differentiation period prior to addition of OSM (or PBS)?

Line 372 ff, why would the effect of E-Cadherin down-regulation be detrimental, it is in fact a normal process upon syncytialization. Moreover, it certainly cannot be reversed. Please rephrase.

Section 3.5, please re-work for several instances of spelling/syntax errors, and try to phrase more compellingly. The repeated comparisons of one treatment versus another are very confusing over time.

Fig. 5C, please double-check the primary data for treatments with (almost) identical replicate values.

Please check entire manuscript for grammar and a few spelling mistakes. Please also reorder the word order when the term “respectively” is being used in a sentence (e.g., “Treatments A and B caused x and y effects, respectively”).

Comments on the Quality of English Language

Just a few minor corrections are needed.

Author Response

Dear reviewer,

Thank you for your involvement and your comments. We've taken the time to respond to each of them and hope our answers will be good for you.

Please, find attached our response document.

Best regards

Reviewer 2 Report

Comments and Suggestions for Authors

The manuscript entitled “Oncostatin M and STAT3 Signalling Pathway Support Human Trophoblast Differentiation by Inhibiting Inflammatory Stress in Response to IFNg and GM-CSF”

Describes the role of OSM and the STAT3 signalling pathway in regulating the biological functions of the trophoblast during inflammation.

The study is well described and interesting just needs some corrections for improvement.

Manuscript authors to improve the work should:

1.     Delete the dot in the title.

2.     Figures 1C-F and 5C show the quantization of wb but write the relative expression on the graph, meaning a qPCR assay. Revise the nomenclatures on the graph in this figure and put Figure 5C instead of 5B. Correct also the graphs in Figure 9 B-C-D-E.

3.     Also, Figures 3F and 8B-C on the graph indicate a PCR quantization shown in figure 8A.

4.     At lane 397, correct “Figures” with “Figure”.

Author Response

Dear reviewer,

Thank you for your involvement and your comments. We have taken the time to respond to each of them and hope our answers will satisfy you.

Please, find attached our response document.

Reviewer 3 Report

Comments and Suggestions for Authors

Here In this manuscript Ravelojaona et al decipher the effect of Oncostatin M on the regulation of signal transduction in response to IFNy-STAT1 and CSF-STAT5 signaling pathways in trophoblast cells. Moreover,  the authors also showed STAT3-dependent and independent effects of OSM for STAT1 and STAT5 signaling with IFNy and GM-CSF stimulation respectively. This is well well-designed study and addresses important questions. However, there are some minor issues.

1) The Lbelling of figure-5 needs to be corrected, Fig- 5C should be 5B and vice versa. 

2) Result section Line 483-487 is not correct. IFNy does not inhibit E-cad expression in Figure 5 B as stated in those lines. 

3) Line 511 figure NO should be 8A. Authors should check the citation of the figure's number in the text carefully. 

 4) The first paragraph of the discussion is more or less a repetition of the introduction section. 

Author Response

Dear reviewer,

Thank you for your involvement and your comments. We have taken the time to respond to each of them and hope our answers will satisfy you.

Please, find attached the document with our responses.

Best regards
